# Pharmacological Blocking of Adiponectin Receptors Induces Alzheimer’s Disease-like Neuropathology and Impairs Hippocampal Function

**DOI:** 10.3390/biomedicines13051056

**Published:** 2025-04-27

**Authors:** Hui-Hui Guo, Hai-Ning Ou, Jia-Sui Yu, Suk-Yu Yau, Hector Wing-Hong Tsang

**Affiliations:** 1Department of Rehabilitation Medicine, The Fifth Affiliated Hospital of Guangzhou Medical University, Guangzhou 510799, China; huihui098@foxmail.com; 2Department of Rehabilitation Medicine, Shaoxing People’s Hospital, Shaoxing 312000, China; 3Department of Rehabilitation, The Second Affiliated Hospital of Guangzhou University of Chinese Medicine, Guangdong Provincial Hospital of Chinese Medicine, Guangzhou 510120, China; haining@gzucm.edu.cn; 4Department of Rehabilitation, Guangdong Provincial Hospital of Chinese Medicine, Guangzhou 510120, China; 5Department of Rehabilitation Sciences, The Hong Kong Polytechnic University, Hong Kong SAR, China; jessejiasui.yu@polyu.edu.hk; 6Mental Health Research Center, The Hong Kong Polytechnic University, Hong Kong SAR, China

**Keywords:** adiponectin, adiponectin receptor antagonist, cognitive impairment, tau, β-amyloid

## Abstract

**Background/Objectives**: Previous studies have shown that adiponectin deficiency or blocking adiponectin receptors (AdipoRs) in the brain can lead to an Alzheimer’s disease (AD)-like neuropathology. While AdipoRs are abundantly expressed in peripheral tissues, the effects of blocking these receptors in the peripheral tissues on the brain are unclear. This study investigates the impacts of blocking AdipoRs with a peripheral administration of ADP400, an antagonist peptide that targets AdipoRs on cognitive performance, hippocampal adult neurogenesis, and AD-like neuropathology in mice. **Methods**: Adult mice were intraperitoneally administered with ADP400 peptide that blocks peripheral AdipoRs continuously for 21 days, followed by a battery of behavioral test for mood and memory performance. **Results**: ADP400-treated mice exhibited impaired memory performance and increased anxiety-like behaviors. Molecular analyses revealed heightened hyperphosphorylation of tau and increased β-amyloid levels, alongside decreased expression of AdipoRs and PP2A in the hippocampus, suggesting a critical role of AdipoRs in AD-like neuropathology. Furthermore, ADP400 treatment significantly reduced hippocampal adult neurogenesis, as indicated by decreased BrdU, Ki67, and DCX staining. Inhibiting peripheral adiponectin receptors could lead to tau hyperphosphorylation and accumulated β-amyloid levels. **Conclusions**: These findings highlight the critical role of peripheral manipulation of adiponectin receptors in modulating cognitive function and adult neurogenesis, offering insights into potential therapeutic strategies for AD and related cognitive disorders.

## 1. Introduction

Alzheimer’s disease (AD) is a progressive neurodegenerative disorder and the leading cause of dementia, posing significant global health challenges [1]. The hallmark features of AD include the formation of extracellular plaques composed of aggregated β-amyloid (Aβ) peptides and the development of intracellular neurofibrillary tangles formed by hyperphosphorylated tau (p-tau) proteins [2]. Increased activity of glycogen synthase kinase 2 (GSK3β) and decreased activity of phosphatase 2A (PP2A) are known to contribute to the development of phosphorylated tau (p-tau) pathology [3,4].

The etiology of AD is complex, influenced by a combination of aging, genetic predispositions, and environmental factors [5]. Substantial evidence indicates a strong association between AD and diabetes mellitus (DM) [6,7], as both conditions share common pathophysiological mechanisms such as neuroinflammation [8], insulin dysfunction [6,9] and impaired glucose metabolism [6,7,9,10]. Notably, depression is linked to cognitive decline and is a significant risk factor for dementia in individuals with diabetes [11]. In summary, AD is characterized by multifaceted and intricate pathologies, with no current preventive measures or cures available [1].

Adiponectin, a hormone derived from adipose tissue, has emerged as a potential therapeutic target in AD due to its diverse physiological roles [12] and involvement in various neurological processes [13,14,15]. It is crucial in regulating glucose and lipid metabolism, insulin sensitivity, and inflammation [16,17]. Altering adiponectin expression could be essential for managing both diabetes and AD. Adiponectin has an important role in AD as found in both clinical and pre-clinical studies [18]. Adiponectin deficiency in adiponectin knockout mice or suppression of adiponectin receptor 1 with the shRNA approach in the brain leads to AD-like neuropathology and memory dysfunction [19,20]. Furthermore, adiponectin deficiency in AD mouse models leads to significant activation of the NLRP3 inflammasome in microglia, whereas overexpression of trimeric adiponectin may suppress microglial-mediated neuroinflammation and improve cognitive performance [21]. Clinical studies have found elevated adiponectin levels in AD patients compared to cognitively normal individuals [22], and lower serum adiponectin and receptor levels are associated with reduced global cognition [23].

Adiponectin interacts with three receptors: AdipoR1, AdipoR2, and T-cadherin. Activation of AdipoR1/AMPK signaling in the brain may regulate neuronal insulin resistance and enhance insulin signaling, potentially ameliorating cognitive deficits in both AD and adiponectin-deficient models [19,24]. Additionally, adiponectin reduces Aβ deposition by modulating AdipoR1/AMPK/NF-κB signaling in the brain [25]. As AD progresses, patients exhibit synaptic dysfunction, neuronal loss, cognitive decline, and memory impairments following the onset of Aβ and tau pathology [26]. Synaptic dysfunction and loss are closely associated with cognitive decline [27]. The low molecular weight form of adiponectin can pass through blood–brain barrier (BBB) to impact brain function. Our previous studies, along with accumulating evidence, suggest that adiponectin significantly impacts the central nervous system, including the modulation of synaptic plasticity [28,29,30,31]. Based on adiponectin’s antidiabetic effects, Okada-Iwabu M et al. developed AdipoRon, an orally active synthetic small-molecule agonist for AdipoR1 and AdipoR2 [32]. Subsequent research has confirmed that AdipoRon can pass through BBB and can protect against the development of AD pathologies, including promoting Aβ clearance [33], inhibiting tau hyperphosphorylation [34], and lowering neurofibrillary tangles (NFTs) [35]. Thus, adiponectin and its receptors represent promising targets for controlling AD through the regulation of neuroinflammation, glucose and lipid metabolism, and insulin sensitivity. Since adiponectin receptors are abundantly expressed in peripheral organs, it is unknown whether disrupting peripheral adiponectin receptors could induce AD-like neuropathology in the brain.

This study focuses on the effects of continuous blockage of peripheral adiponectin receptors on cognitive function. We investigated changes in Aβ and p-tau levels, hippocampal adult neurogenesis, as well as cognitive learning and memory performance. By elucidating the link between peripheral adiponectin receptor function and AD-associated cognitive impairment, the study provides valuable insights into the possible influence of peripheral adiponectin receptors on hippocampal plasticity and the critical role of maintaining functions of intact adiponectin signaling as a preventative measure for the pathogenesis of AD.

## 2. Materials and Methods

### 2.1. Animals

All animal experiments were conducted in accordance with the Institutional Animal Care and Use Committee (IACUC) Guidelines on the Use of Laboratory Animals and were approved by the Hong Kong Polytechnic University (PolyU) Shenzhen Research Institute Committee on Animal Care.

Adult male wild-type (WT) C57BL/6J mice, aged 6 weeks, were obtained from the Hong Kong Polytechnic University (PolyU) Centralised Animal Facility (Shenzhen) on 7 January 2022 (ASESC Case No.: 20-21/303-RS-R-NSFC). The mice were housed in a standard animal holding facility with controlled ambient temperature (22 ± 2 °C) and humidity (40–60%), maintained on a 12-h light/dark cycle, with food and water available ad libitum.

### 2.2. Drug Administration

The ADP400 peptide, an adiponectin receptor antagonist, was procured from Sangon Biotech Co., Ltd. (Shanghai, China). ADP400, as a synthesized peptide acting as an antagonist to AdipoRs, has limited biodistribution, primarily targeting organs involved in peptide elimination following intraperitoneal (i.p.) injection. This characteristic suggests that ADP400 may predominantly affect peripheral tissues rather than the central nervous system, due to its restricted distribution [36]. Mice received i.p. administration of the ADP400 peptide at a dosage of 0.5–1 mg/kg/day [36] or a vehicle as a control treatment continuously for 21 days. To assess the survival of newly formed cells, mice were injected with bromodeoxyuridine (BrdU, Yeasen Biotechnology (Shanghai) Co., Ltd. Shanghai, China) i.p. at a dose of 50 mg/kg, dissolved in 0.9% saline 7 days prior to the initiation of ADP400 peptide treatment (see the Figure 1 below).

### 2.3. Behavioral Tests

#### 2.3.1. Open Field Test (OFT)

To assess anxiety-like behavior and locomotor activity, mice were allowed to explore a clean open field (dimensions: 50 × 50 × 50 cm) for 10 min under dim lighting, as previously described [28]. During the first 5 min of the test, we analyzed the total distance traveled (in cm), mean velocity (in cm/s), and time spent as well as the frequency of entries and time spent in the central area using EthoVision XT15 software (Noldus, Wageningen, The Netherlands).

#### 2.3.2. Y-Maze Task

The hippocampal-dependent spatial memory was evaluated using the Y-maze task, as previously described [29]. In the pre-training phase, mice were allowed to freely explore the starting arm and familiar arm (dimensions: 30 × 6 × 8 cm) of the Y-maze for 5 min, after which they were returned to their home cage for four hours. Subsequently, the mice were reintroduced to the maze with access to the previously blocked novel arm for 5 min. During this time, the duration spent and the number of visits to each arm were recorded over a 5-min period and analyzed using EthoVision XT15 software (Noldus, Wageningen, The Netherlands). The percentage of time or visits was calculated using the following formula: % time/visit = (exploration time or entries in the novel arm/total number of arm entries or total time) × 100.

#### 2.3.3. Novel Object Recognition Test

The novel object recognition (NOR) test was employed to evaluate working memory performance, following established protocols [31]. On day 1, mice were allowed to habituate in an open field test apparatus (dimensions: 50 × 50 × 50 cm) for 15 min. On day 2, the mice explored two identical objects within the testing apparatus for 10 min. After a two-hour interval, the mice were reintroduced to the apparatus and allowed to explore one novel object and one familiar object for 5 min. The time spent exploring the familiar (F) and novel (N) objects was recorded. The exploration index was calculated using the following formula: (N − F)/(N + F).

#### 2.3.4. Elevated Plus Maze Test

Anxiety-like behavior was assessed using the elevated plus maze (EPM) test [37]. In this setup, mice were placed at the center of a maze consisting of two open arms (30 × 6 cm) and two closed arms (30 × 7 × 8 cm), elevated 50 cm above the floor. The animals were allowed to explore the EPM for 5 min. The time spent in each arm and the number of entries into the arms were recorded and analyzed using Noldus EthoVision XT15 software (Noldus, Wageningen, The Netherlands).

#### 2.3.5. Forced Swim Test

The forced swim test (FST) was employed to assess depression-like behavior, following established protocols [28]. Mice were placed in a cylinder (height: 30 cm; diameter: 15 cm) filled with water maintained at 23  ±  2 °C, and their activity was recorded for 6 min using a camera. The immobility time during the last 4 min was calculated by a trained researcher who was blinded to the sample identities.

### 2.4. Tissue Preparation for Histochemistry

Mice were anesthetized using isoflurane, and blood samples were collected prior to perfusion with 0.9% saline, followed by 4% paraformaldehyde. The brain tissues were then isolated and post-fixed overnight at 4 °C. Subsequently, the tissues were transferred to a 30% sucrose solution until they sank. Coronal sections (1-in-6 series, 30 μm thickness) were prepared using a vibratome (Leica Biosystems, Nussloch, Germany) and stored in an antifreeze cryoprotectant solution consisting of 30% glycerol and 30% ethylene glycol at 4 °C until further use.

### 2.5. Immunohistochemistry Staining

Immunohistochemistry staining was performed as previously described [28,29]. After antigen retrieval in citric acid buffer (pH 6.0) and washing with PBS, sections were incubated with primary antibodies overnight. The following primary antibodies were used: rabbit anti-Ki-67 (1:1000, #9449, Cell Signaling Technology, Danvers, MA, USA), mouse anti-DCX (1:200, sc-271390, Santa Cruz Biotechnology, Dallas, TX, USA), and mouse anti-BrdU (1:500, #5292, Cell Signaling Technology, USA). This was followed by incubation with secondary antibodies, including anti-mouse antibodies (1:200, TI-2000, Vector Laboratories, Newark, CA, USA) and anti-rabbit antibodies (1:200, TI-1000, Vector Laboratories, CA, USA), for 2 h at room temperature. Finally, the sections were cover-slipped with an anti-fade mounting medium containing DAPI (Cat No. P0131; Beyotime), and immunofluorescence signals were observed using a Nikon AXE laser confocal microscope (Nikon, Tokyo, Japan).

### 2.6. Quantification of BrdU, Ki67, and DCX Positive Cells

In brief, BrdU, Ki67, and DCX positive cells were counted in sections ranging from bregma −1.30 to −3.80 mm [38]. Only positive cells located in the dentate gyrus (DG) sub-granular zone and granular cell layer were included in the count, while those in the uppermost focal plane were excluded [38]. The sub-granular zone is defined as the area within approximately two cell bodies (~20 μm) from the inner edge of the molecular layer [38]. Quantification was conducted in a sample-blinded manner to minimize observer bias and ensure reliability.

### 2.7. Western Blot Analysis

Hippocampal tissues were lysed using RIPA buffer (Beyotime, Shanghai, China) supplemented with protease and phosphatase inhibitors (Thermo Scientific, Waltham, MA, USA). Total protein concentrations were determined with the BCA Protein Assay Kit (Thermo Fisher Scientific, Inc., Waltham, MA, USA). Proteins extracted from brain tissue (30 µg) were subjected to SDS-PAGE and subsequently transferred to PVDF membranes. The membranes were blocked with 5% BSA (Macklin, Shanghai, China) and incubated overnight at 4 °C with primary antibodies, including p-Tau S404 (1:1000, Abcam, Cambridge, MA, USA, #ab92676), p-Tau S396 (1:1000, Abcam, #ab109390), β-actin (1:1000, Cell Signaling Technology, Danvers, MA, USA,), PP2Ac (1:1000, Cell Signaling Technology, Danvers, MA, USA), GSK-3β (1:1000, Cell Signaling Technology, Danvers, MA, USA), GAPDH (1:5000, Genetex, Irvine, CA, USA, #GTX100118), α-tubulin (1:1000, Invitrogen™, Waltham, MA, USA, #62204), and β-amyloid (1:1000, Cell Signaling Technology, Danvers, MA, USA). The membranes were then incubated with secondary antibodies: anti-mouse (1:2000, Cell Signaling Technology, Danvers, MA, USA) and anti-rabbit (1:2000, Cell Signaling Technology, Danvers, MA, USA) for 2 h at room temperature. Immunoreactive bands were visualized and quantified using the ChemiDoc Imaging System (Bio-Rad, Hercules, CA, USA).

### 2.8. Quantitative Reverse Transcription PCR

The total RNA was extracted using the Total RNA Kit (R6934-02, Omega, Macon, GA, USA). The extracted RNA was then reverse-transcribed into cDNA using the Hifair^®^ AdvanceFast 1st Strand cDNA Synthesis Kit (11149ES60, Yeasen, Shanghai, China). Target genes were amplified with a PCR thermal cycler (Thermo Fisher Scientific, USA) and SYBR Green (11203ES08, Yeasen, Shanghai, China) following the manufacturer’s protocol. The sequences of the primers used are listed in Table 1.

### 2.9. Statistical Analysis

Parametric data were analyzed using GraphPad Prism 9 software (GraphPad Software, San Diego, CA, USA). Differences between the two groups were assessed using the Student’s *t*-test or the Kolmogorov–Smirnov test, as appropriate. Data are presented as mean ± standard error of the mean (SEM), with <0.05 considered statistically significant.

## 3. Results

### 3.1. Blocking Adiponectin Receptors Downregulates Adiponectin Receptor Expression in the Hippocampus

Studies have suggested that AdipoRs are predominantly present in the hippocampus and play a significant role in mediating adiponectin signaling, which influences cognitive function [14,28,29,30,39]. To investigate this further, we blocked adiponectin signaling by administering adiponectin receptor antagonist peptides (ADP400) [36]. Our results demonstrated a significant decrease in the levels of AdipoR1 (Figure 2A; two-sample Kolmogorov–Smirnov test: D = 1.000, *p* = 0.0286) and AdipoR2 (Figure 2B; two-sample *t*-test: *t*(6) = 2.672, *p* = 0.0173) in the hippocampus. Additionally, there was a noticeable trend toward reduced expression of PP2A mRNA (Figure 2C; two-sample *t*-test: *t*(6) = 2.428, *p* = 0.0513) and GSK-3β mRNA (Figure 2D; two-sample *t*-test: *t*(6) = 1.576, *p* = 0.1661) in the ADP400-treated group compared to the control group, although these differences did not reach statistical significance. Therefore, peptide ADP400 serves as a validated tool for studying adiponectin functions in the hippocampus.

### 3.2. Blocking Adiponectin Receptors Induces Tau Hyperphosphorylation and Increases β-Amyloid Levels in the Hippocampus

To further explore the impact of ADP400 on cognitive function, we examined hippocampal levels of hyperphosphorylated tau protein and Aβ. Western blotting assays revealed significant hyperphosphorylation of tau at specific sites (Serine 396 and Serine 404) in the ADP400-treated group (Figure 3A,B; Figure 3A: two-sample *t*-test, *t*(15) = 2.226, *p* = 0.0418; Figure 3B: two-sample *t*-test, *t*(16) = 3.959, *p* = 0.0011). Consistent with the observed tau pathology in the hippocampus, the ADP400 peptide significantly increased β-amyloid protein expression levels (Figure 3C; two-sample *t*-test, *t*(8) = 2.534, *p* = 0.0350). Additionally, we found that the ADP400 peptide significantly reduced PP2A protein expression (Figure 3D; two-sample *t*-test, *t*(6) = 2.640, *p* = 0.0386), while no significant difference was observed in the protein kinase GSK-3β levels (Figure 3E; two-sample *t*-test, *t*(8) = 2.213, *p* = 0.0578). These results suggest that the adiponectin signaling pathway plays a crucial role in the progression of both Aβ and tau pathologies.

### 3.3. Blocking Adiponectin Receptors Induces Anxiety-like Behavior

To characterize the effects of adiponectin signaling deficiency on cognitive deficits, we evaluated anxiety-like behavior and locomotor function in mice. Compared to the control group, ADP400-treated mice exhibited higher anxiety levels in the forced swim test (Figure 4A; two-sample *t*-test: *t*(18) = 3.061, *p* = 0.0067, and Figure 4E) Additionally, we observed an increase in anxiolytic-like behavior in the elevated plus maze (Figure 4B; two-sample Kolmogorov–Smirnov test: D = 0.6869, *p* = 0.0187, and Figure 4E) and the open field test (Figure 4C; two-sample *t*-test: *t*(18) = 2.404, *p* = 0.0272, and Figure 4E). However, no significant differences were found between the control and ADP400-treated mice in locomotor activity (Figure 4D; two-sample *t*-test: *t*(18) = 1.860, *p* = 0.0793). These results suggest that inhibition of adiponectin receptors by the ADP400 peptide leads to depression- and anxiety-like behavior.

### 3.4. Blocking Adiponectin Receptors Impairs Cognitive Function

To elucidate the effects of impaired adiponectin signaling on cognitive deficits, we assessed learning and memory functions in mice. The novel object recognition tests revealed that treatment with the ADP400 peptide significantly induced cognitive deficits (Figure 5A; two-sample Kolmogorov–Smirnov test: D = 0.8182, *p* = 0.0026; Figure 5B; two-sample *t*-test: *t*(18) = 2.301, *p* = 0.0336; and Figure 5E). Consistent with these findings, results from the Y-maze test indicated that ADP400-treated mice showed less interest in exploring novel arms compared to control mice (Figure 5C; two-sample *t*-test: *t*(18) = 2.167, *p* = 0.0439; and Figure 5E), indicating impaired spatial memory. Although there was no significant difference, a declining trend was observed in the number of entries into the novel arms (Figure 5D; two-sample Kolmogorov–Smirnov test: D = 0.3434, *p* = 0.6035).

### 3.5. Blocking Adiponectin Receptors Reduces Hippocampal Neurogenesis

Next, we investigated the relationship between ADP400 peptide treatment and hippocampal neurogenesis, using BrdU, Ki67, and DCX staining as markers (Figure 6). The survival of newly formed cells labeled with BrdU was significantly reduced by ADP400 peptide treatment (Figure 6A; two-sample *t*-test: *t*(10) = 2.381, *p* = 0.0386). Similarly, the number of immature neurons, indicated by DCX+ cells, was reduced in ADP400-treated mice (Figure 6B; two-sample *t*-test: *t*(10) = 2.558, *p* = 0.0285). Additionally, ADP400 peptide treatment significantly reduced hippocampal cell proliferation (Figure 6C; two-sample *t*-test: *t*(10) = 3.172, *p* = 0.0100). These findings suggest that the ADP400 peptide inhibits adult hippocampal neurogenesis.

## 4. Conclusions

In this study, we examined the critical role of peripheral adiponectin receptors on AD neuropathology and hippocampal function by using the adiponectin receptor antagonist ADP400 peptide. Our findings revealed that the inhibition of peripheral adiponectin receptors increased tau hyperphosphorylation, elevated β-amyloid expression levels in the hippocampus in concurrent with increased anxiety-like behavior, impaired cognitive memory performance, and reduced hippocampal neurogenesis in adult mice. These results highlight the potential implications of disrupted peripheral adiponectin receptors in contributing to cognitive deficits and neurodegenerative processes.

We found that the inhibition of the adiponectin receptors contributes to cognitive dysfunction. Adiponectin signaling is known to play a crucial role in synaptic plasticity [28,29,30,39] and neuroprotection [15,40]. The suppression of peripheral adiponectin signaling observed in our study may have contributed to these outcomes. Specifically, blocking adiponectin receptors with a peripheral administration of ADP peptide induced tau hyperphosphorylation and altered β-amyloid levels in the hippocampus, which are key pathological features associated with neurodegenerative disorders (Figure 3), suggesting a critical role of intact adiponectin receptors in maintaining physiological levels of tau phosphorylation and altered β-amyloid production. The relationship between tau hyperphosphorylation and β-amyloid accumulation is of particular interest in AD research. Tau hyperphosphorylation is known to disrupt microtubule stability and promote the formation of neurofibrillary tangles (NFTs) [41]. Within the context of AD pathology, the accumulation of Aβ plaques is considered an early precipitating event, while the development of tau NFTs is seen as a more immediate trigger of subsequent neuronal impairment and degeneration [42,43]. The increased β-amyloid levels observed in our study suggest a potential interaction between peripheral adiponectin signaling and hyperphosphorylated tau and β-amyloid deposition (Figure 3).

Additionally, studies have identified PP2A and GSK3β as key enzymes involved in the modification of tau phosphorylation [44]. Our results demonstrated a decreasing trend in both PP2A and GSK3β mRNA expression in ADP400-treated mice. PP2A and GSK3β play an important role in regulating phosphorylated tau and cognitive impairments. Although the underlying mechanisms of adiponectin’s action on modulating AD neuropathology remain largely unclear, its signaling pathways can influence key proteins and enzymes involved in AD. Emerging findings suggest that central adiponectin signaling might modulate p-tau accumulation through direct action on tau phosphorylation or indirect interactions with GSK3β and the enzyme PP2A. PP2A dephosphorylates tau at several phosphorylation sites. GSK3β is a crucial kinase that phosphorylates tau, contributing to the accumulation of p-tau. Adiponectin signaling can inhibit GSK3β activity [4]. It attenuates tau hyperphosphorylation at multiple AD-related sites by activating Ser9-phosphorylated GSK3β with increased Akt and PI3K activity [45]. Another study showed that adiponectin blocks the phosphorylation of the GSK-3 β/β-catenin pathway [46]. Additionally, adiponectin may enhance the activity of PP2A, a major phosphatase that dephosphorylates tau. Our unpublished data indicate that adiponectin knockout reduces PP2A activity and increases tau phosphorylation. Protein Phosphatase Methyl Esterase-1 (PME-1) is an enzyme that demethylates the catalytic subunit of PP2A, directly regulating its activity. This suggests that PME-1 may play a role in mediating the effects of adiponectin on the dephosphorylation of tau proteins [47,48]. Overall, adiponectin signaling may exert neuroprotective effects by modulating these pathways, potentially offering therapeutic benefits in AD. However, the exact mechanisms and the extent of these effects are still under investigation, and more research is needed to fully understand these interactions.

Our study uncovered a significant effect of ADP400 peptide with peripheral administration on increasing depression/anxiety-like behavior and impairing memory performance, concurrent with aberrant adult hippocampal neurogenesis in mice. Selective deletion of adiponectin receptor 1 in the dorsal raphe nucleus specifically could lead to depression-like behavior [49], and specific knockdown of AdipoR1 with siRNA in the brain is shown to be critical for restoring memory impairment and AD-like neuropathology [24]. Previous research has emphasized the central role of adiponectin and its receptors present in the brain in modulating anxiety and stress responses [15,50,51]. Here, we demonstrated that blocking peripheral adiponectin receptors will also induce AD-like neuropathology. Notably, chronic treatment with i.p. administration of ADP400 reduced the mRNA expression of AdipoR1 and AdipoR2 in the hippocampus, suggesting that a decrease in adiponectin action in the brain could be affected by changes in peripheral adiponectin receptor functions. There are several possible explanations for the reduced mRNA expression levels of adiponectin receptors caused by ADP400. First, this reduction could lead to decreased sensitivity or responsiveness to adiponectin, potentially resulting in lower adiponectin levels as a compensatory mechanism. Second, ADP400 might directly affect the function of adipose tissue or other organs with abundant adiponectin receptor expression, altering overall adiponectin secretion or action. This could occur through changes in the expression of genes involved in the synthesis or release of adiponectin or affecting adiponectin signaling directly.

Blocking of peripheral adiponectin receptors with ADP400 peptide could lead to significant metabolism-related effects, as adiponectin plays a crucial role in regulating metabolic processes [12]. Adiponectin enhances insulin sensitivity, so blocking its receptors (AdipoRs) could reduce insulin sensitivity, potentially contributing to insulin resistance and increasing the risk of type 2 diabetes. Disruption of peripheral adiponectin signaling might impair glucose uptake and utilization, leading to elevated blood glucose levels. Additionally, adiponectin influences lipid metabolism by promoting fatty acid oxidation and reducing triglyceride levels. Blocking peripheral AdipoRs could result in dyslipidemia, characterized by increased triglycerides and altered lipid profiles. Chronic disruption of adiponectin signaling through the AdipoR blockade can have significant effects on metabolism and related pathways, potentially contributing to various metabolic and cardiovascular disorders. We observed a significant reduction in AdipoRs expression in the hippocampal region after chronic treatment with ADP400. Further research is needed to fully understand the metabolic implications of ADP400 treatment and the details mechanisms by which peripheral adiponectin receptor disruption affects adiponectin-related metabolic function in the brain and hence contributes to AD-like pathology in the hippocampus.

In addition to cognitive dysfunction, we observed a decrease in hippocampal neurogenesis following ADP400 peptide administration. Impaired neurogenesis is associated with various neurological disorders and may contribute to hippocampal dysfunction. The reduction in neurogenesis underscores the potential impact of disrupted adiponectin signaling on hippocampal integrity and function. Adiponectin, an adipokine secreted by adipose tissue, has been shown to influence neurogenesis in the hippocampus [28,52]. Studies indicate that adiponectin promotes the proliferation and differentiation of neural progenitor cells in the hippocampus. Adiponectin enhances hippocampal neurogenesis and synaptic plasticity [28,52]. The presence of adiponectin receptors, AdipoR1 and AdipoR2, in the hippocampus suggests a direct role in modulating neurogenic pathways [53,54]. Liver gene delivery to overexpress trimeric adiponectin suppresses the neuroinflammasome, consequently improves learning and memory, and reduces amyloid load, suggesting a potential therapy for AD by enhancing adiponectin signaling [21]. This effect could be mediated by AdipoR1 [25,32]. The expression of adiponectin receptors in the brain, particularly in regions like the hippocampus, underscores their importance in mediating the neuroprotective effects of adiponectin. Activation of adiponectin signaling by its receptor agonist AdipoRon (which can pass through blood–brain barrier) significantly enhances hippocampal neuroplasticity, including neurogenesis, dendritic complexity, and synaptic plasticity. Similarly, activation of adiponectin receptors can mitigate neuroinflammatory responses and promote neuronal survival, further supporting their role in cognitive health [21,25,55]. Conversely, adiponectin deficiency can lead to reduced neurogenesis, cognitive impairment, and AD neuropathology [19,56]. Our results further confirmed that blocking adiponectin signaling results in cognitive impairments in association with enhanced AD neuropathology and hippocampal impairment.

Our results demonstrated that chronic intraperitoneal (i.p.) administration of ADP400 downregulated adiponectin receptor expression in the hippocampus. Adiponectin receptors are also present in other tissues, including the liver, muscles, heart, adipose tissue, pancreas, and kidneys. There is currently a lack of evidence demonstrating that ADP400 can cross the blood–brain barrier and enter the brain; we cannot rule out the possibility that the effects of i.p. administration of ADP400 are linked to changes in peripheral adiponectin secretion or peripheral inflammatory responses, which may lead to alterations in neuroinflammation. This suggests that the observed effects might not be limited to the downregulation of adiponectin receptor-mediated signaling in the brain.

The study presents intriguing findings on the effects of chronic intraperitoneal administration of ADP400 on adiponectin receptor expression in the hippocampus. However, several experimental limitations should be considered. The ADP400 peptide is a large peptide that generally has difficulty crossing the BBB due to its selective permeability. However, whether the observed effects are directly due to central nervous system interactions or are secondary to peripheral changes warrants investigation to further confirm. The systemic administration of ADP400 could influence peripheral adiponectin levels or inflammatory responses, which might indirectly affect hippocampal function and neuroinflammation. Additionally, the study does not account for potential compensatory mechanisms in other tissues where adiponectin receptors are present, and future studies focusing on ADP400-triggered peripheral response leading to hippocampal changes warrant further investigation.

In conclusion, our study provides evidence for the detrimental effects of blocking adiponectin receptors by i.p. administration with an adiponectin receptor antagonist, highlighting its negative effects in interrupting adiponectin receptors in the hippocampus and its impact on tau hyperphosphorylation, anxiety-like behavior, cognitive function, and hippocampal neurogenesis. These findings enhance our understanding of the role of adiponectin receptors in hippocampal function and their potential involvement in neurodegenerative processes. Our findings have suggested a novel role of peripheral manipulation of adiponectin receptors in regulating hippocampal structural plasticity and cognitive memory performance. Further investigations are warranted to explore potential therapeutic interventions that restore adiponectin receptor-mediated signaling in the diseased condition to reduce associated AD neuropathology.

## Figures and Tables

**Figure 1 biomedicines-13-01056-f001:**
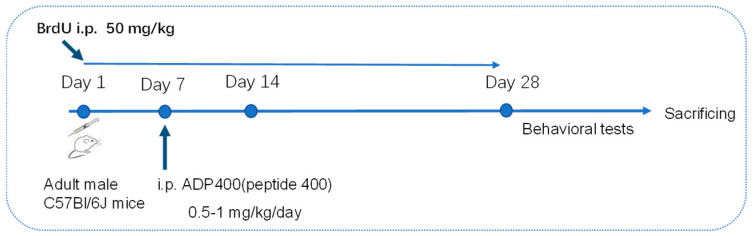
Timeline of ADP400 treatment, BrdU injection and behavioral tests.

**Figure 2 biomedicines-13-01056-f002:**
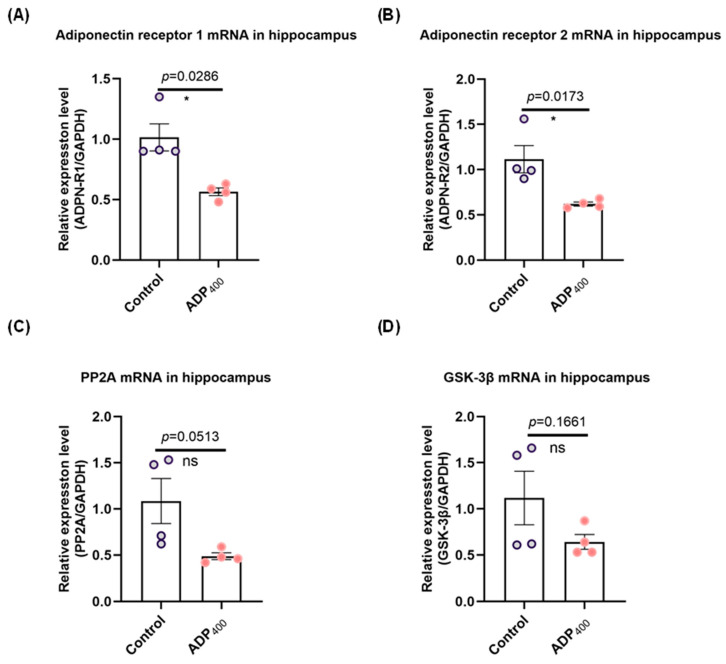
The effects of the ADP400 peptide on adiponectin, its receptors, and the expression levels of protein phosphatases PP2A and protein kinases GSK-3β mRNA were assessed. (**A**–**D**) The expression levels of AdipoR1, AdipoR2, PP2A, and GSK-3β mRNA were quantified using RT-qPCR, with n = 4 mice per group. Data were presented as mean ± standard error of the mean (SEM). Differences between the two groups were analyzed using the Student’s *t*-test or the Kolmogorov–Smirnov test, as appropriate. * *p* < 0.05 compared to the control group. “ns” means non-significant difference.

**Figure 3 biomedicines-13-01056-f003:**
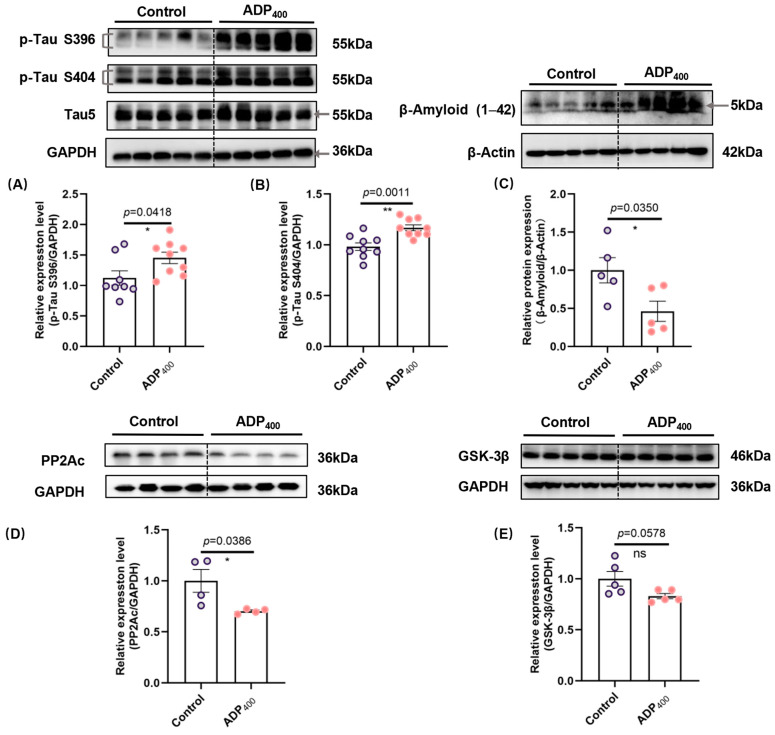
Inhibition of adiponectin receptors leads to increased aggregation of β-amyloid and tau pathology. (**A**) Quantification of phosphorylated tau at Serine 396 (p-Tau S396), (**B**) phosphorylated tau at Serine 404 (p-Tau S404), and (**C**) β-amyloid in the hippocampus. (**D**,**E**) Quantification of GSK-3β and PP2A. n = 4–9 mice per group. Data are presented as mean ± SEM. Differences between the two groups were analyzed using the Student’s *t*-test or the Kolmogorov–Smirnov test, as appropriate. * *p* < 0.05, ** *p* < 0.01 compared to Control. “ns” means non-significant difference.

**Figure 4 biomedicines-13-01056-f004:**
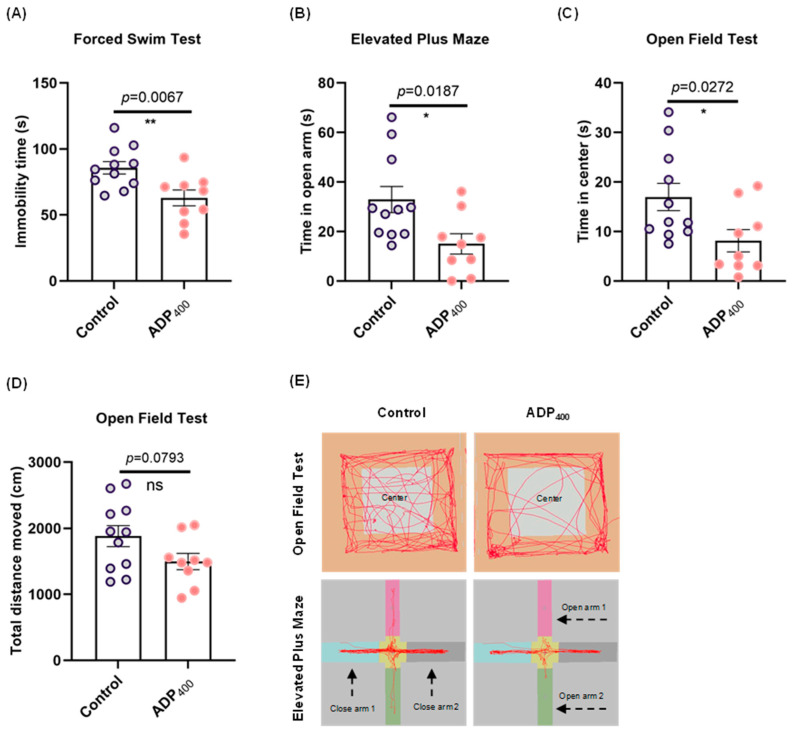
Effects of inhibited adiponectin receptors on depression-like behavior. (**A**) Immobility time in the forced swim test (FST). (**B**) Exploration time in the open arms of the elevated plus maze test (EPT). (**C**) Time spent in the center of the open field test (OFT). (**D**) Total movement in the OFT. (**E**) Representative traces of animal behavioral performance in the OFT and EPT. Each group consisted of 9 to 11 mice. Data are presented as mean ± SEM. Differences between the two groups were analyzed using the Student’s *t*-test or the Kolmogorov–Smirnov test, as appropriate. * *p* < 0.05, ** *p* < 0.01 compared to the control group. “ns” means non-significant difference.

**Figure 5 biomedicines-13-01056-f005:**
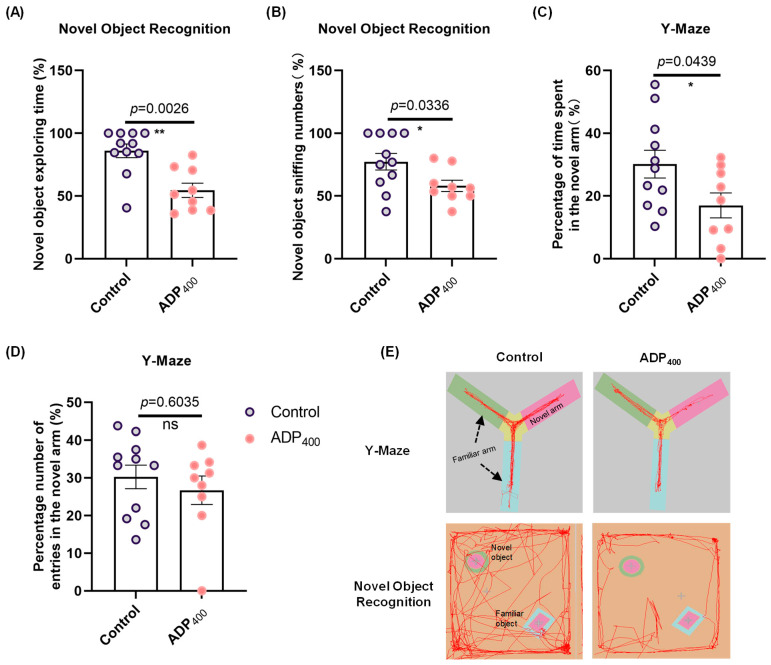
Behavioral phenotypes of mice with disrupted adiponectin receptor function in memory tasks. (**A**) Percentage of time spent and (**B**) number of sniffing events directed at the novel object in the Novel Object Recognition (NOR) test. (**C**) Percentage of time spent and (**D**) number of visits to the novel arm in the Y-maze test. (**E**) Representative traces of animal behavioral performance in the NOR and Y-Maze tests. Sample size: n = 11 or 12 mice per group. Data are presented as mean ± SEM. Differences between the two groups were analyzed using the Student’s *t*-test or the Kolmogorov–Smirnov test, as appropriate. * *p* < 0.05, ** *p* < 0.01 compared to Control. “ns” means non-significant difference.

**Figure 6 biomedicines-13-01056-f006:**
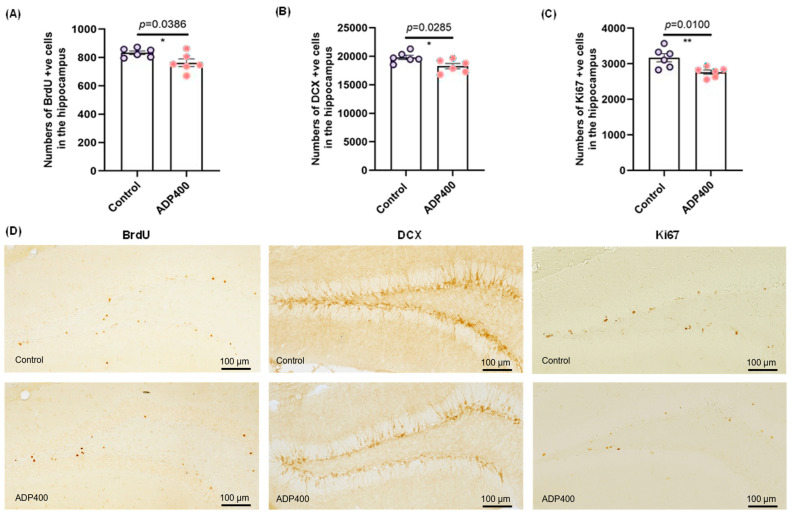
Inhibition of adiponectin receptors by ADP400 peptide reduces adult hippocampal neurogenesis. (**A**) Quantification of BrdU+ surviving adult-born cells. (**B**) Doublecortin (DCX)+ immature neurons in the hippocampal dentate gyrus. (**C**) Ki67+ proliferating cells. (**D**) Representative images of BrdU, Ki67, and DCX immunostaining. Scale bar, 200×. n = 6 mice per group. Data are presented as mean ± SEM. Differences between the two groups were analyzed using the Student’s *t*-test or the Kolmogorov–Smirnov test, as appropriate. * *p* < 0.05, ** *p* < 0.01.

**Table 1 biomedicines-13-01056-t001:** List of primers for RT-qPCR.

Gene Name	Primer (5’ to 3’)
ADNR1-F	GAAAGACAACGACTACCTGCTAC
ADNR1-R	CGTCAAGATTCCCAGAAAGAG
ADNR2-F	CCACCATAGGGCAGATAGG
ADNR2-R	TGAACAAAGGCACCAGCAA
Ppp2ca-F	ATGGA CGAGA AGTTG TTCAC C
Ppp2ca-R	CAGTG ACTGG ACATC GAACC T
GSK-3β-F	CATCCTTATCCCTCCTCACGCT
GSK-3β-R	TATTGGTCTGTCCACGGTCTCC
GAPDH-F	TTCCTACCCCCAATGTATCCG
GAPDH-R	CATGAGGTCCACCACCCTGTT

## Data Availability

Data will be made available by the authors on request.

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
