# Peer review of "Pharmacological Blocking of Adiponectin Receptors Induces Alzheimer’s Disease-like Neuropathology and Impairs Hippocampal Function"

_biomedicines, 2025, doi:10.3390/biomedicines13051056_

Round 1
Reviewer 1 Report (Previous Reviewer 3)
Comments and Suggestions for Authors
Dear Editor,
Thank you for the revised version of the manuscript titled "Pharmacological Blocking of Adiponectin Receptors Induces Alzheimer’s Disease-Like Neuropathology and Impairs Hippocampal Function" by H. Gou et al., submitted to Biomedicines.
The authors have adequately addressed my previous concern regarding Figure 5 (DCX staining). They have updated the figure with a new image for the ADP 400-treated group, and the data now appear convincing.
I have no further major comments or concerns, aside from a minor suggestion to check for typographical errors (e.g., line 398).
The manuscript is suitable for acceptance in its current form.
Thank you.
Reviewer 2 Report (Previous Reviewer 2)
Comments and Suggestions for Authors
I have read the response letter and the revised version. From my point of view, the quality has improved a bit.
Reviewer 3 Report (Previous Reviewer 1)
Comments and Suggestions for Authors
Authors have addressed my concerns in the revised manuscript
This manuscript is a resubmission of an earlier submission. The following is a list of the peer review reports and author responses from that submission.
Round 1
Reviewer 1 Report
Comments and Suggestions for Authors
In this paper, authors investigate if inhibition of adiponectin signalling (via intraperitoneal administration of ADP400, a peptide antagonist of adiponectic receptor (AdipoR) 1 & 2) leads to Alzheimer disease (AD) like phenotype. After injection of ADP400 for 21 days, authors perform western blot & qPCR to check downstream signaling tartegs of adiponectin signaling and also perform behavioural studies to assess the locomotar activity, memory test, anxiety & depression-like behaviour, etc.
Authors show inhibition of Adiponectin signaling by ADP400 leads to decrease in AdipoR1 and R2 mRNA levels along with its downstream targets PP2A and GSK-3b mRNAs. Notably, ADP400 administration also increased Tau hyperphosphorylation and led to accumulation of b-amyloid protein. Further, ADP400 injected animals showed increased anxiety and depression with a concomitant decrease in cognitive function, without an overt change in locomoter activity. Mechanistically, administration of ADP400 led to reduction in hippocampal neurogenesis evidenced by decreased neuronal proliferation as well as decreased number of immature neurons. Thus, this paper provides critical evidences linking adiponectin signaling with AD and possibly other neurodegenerative disorders. Some comments are provided which needs to be addressed before publication.
- AdipoR1 and AdipoR2 receptors are also expressed in other tissues such as liver, muscle, heart, adipose tissue, pancrease and kidney. How does the authors rule out the systemic effect of ADP400 on neuroinflammation, especially given that ADP400 is administered i.p but not directly into the brain? This can be discussed as a limitation of this paper.
- Does ADP400 penetrate blood brain barrier? Are there any prior literature to support this?
- How was the dose for ADP400 chosen for animal injection?
- Intro should discuss how GSK3b and PP2A are linked to adiponectin signaling so that we can relate this to, why these two genes are actually tested in ADP400 injected mice.
- These peptides are receptor antagonist for adiponectin receptor which works at the protein level to inhibit the receptor. But authors show an transcriptional decrease in Adiponectin receptor 1 & 2 mRNA. Can the author provide an explanation on how inhibition of Adiponectin receptor protein leads to inhibition of its mRNA?
- It has earlier been shown that adiponectin promotes the inhibitory phosphorylation of GSK3B at Ser 9 leading to suppression of Tau hyperproliferation. Given that Tau hyperphosphorylation is significantly high in ADP400, authors could have tested GSK3B ser 9 phosphorylation status instead of the total protein.
- Authors can cite PMID 38280832, which is a recent review summarizing the role of adiponectin in AD in both preclinical and clinical studies
Author Response
Response to comments from Reviewer 1
- AdipoR1 and AdipoR2 receptors are also expressed in other tissues such as liver, muscle, heart, adipose tissue, pancrease and kidney. How does the authors rule out the systemic effect of ADP400 on neuroinflammation, especially given that ADP400 is administered i.p but not directly into the brain? This can be discussed as a limitation of this paper.
- Does ADP400 penetrate blood brain barrier? Are there any prior literature to support this?
Response to Q1-Q2: Unfortunately, no literature is found about this. We mentioned this in line 314-315.
We have added in a paragraph stating this point as experimental limitation, as below: line 359-365
“Our results demonstrated that chronic intraperitoneal (i.p.) administration of ADP400 downregulated adiponectin receptor expression in the hippocampus. Adiponectin receptors are also present in other tissues, including the liver, muscles, heart, adipose tissue, pancreas, and kidneys. There is currently a lack of evidence demonstrating that ADP400 can cross the blood-brain barrier and enter the brain. Therefore, we cannot rule out the possibility that the effects of i.p. administration of ADP400 are linked to changes in peripheral adiponectin secretion or peripheral inflammatory responses, which may lead to alterations in neuroinflammation. This suggests that the observed effects might not be limited to the downregulation of adiponectin receptor-mediated signaling in the brain”
3.How was the dose for ADP400 chosen for animal injection?
Response: The dose for ADP400 was chosen based on the paper showing its effective effects in blocking adiponectin receptor activity by Laszlo et al. The reference is listed in the maternal and method section 2.2 Drug administration.
Reference:
Otvos, L.; Knappe, D.; Hoffmann, R.; Kovalszky, I.; Olah, J.; Hewitson, T.D.; Stawikowska, R.; Stawikowski, M.; Cudic, P.; Lin, F.; et al. Development of Second Generation Peptides Modulating Cellular Adiponectin Receptor Responses. Front Chem 2014, 2, 93, doi:10.3389/fchem.2014.00093.”
4. Intro should discuss how GSK3b and PP2A are linked to adiponectin signaling so that we can relate this to, why these two genes are actually tested in ADP400 injected mice.
Responses: We have added background information as suggested in introduction. Line 38-40 as below:
“Increased activity of glycogen synthase kinase 3 beta (GSK3β) and decreased activity of protein phosphatase 2A (PP2A) are known to contribute to the development of phosphorylated tau (p-tau) pathology”
5. These peptides are receptor antagonist for adiponectin receptor which works at the protein level to inhibit the receptor. But authors show an transcriptional decrease in Adiponectin receptor 1 & 2 mRNA. Can the author provide an explanation on how inhibition of Adiponectin receptor protein leads to inhibition of its mRNA?
Response: We added in the contents to explain the downregulation of AdipoRs by ADP400 as below: Line 321-327.
“Notably, chronic treatment with ADP400 reduced the mRNA expression of AdipoR1 and AdipoR2 in the hippocampus, suggesting a decrease in adiponectin action in this region. There are several possible explanations for the reduced mRNA expression levels of adiponectin receptors caused by ADP400. First, this reduction could lead to decreased sensitivity or responsiveness to adiponectin, potentially resulting in lower adiponectin levels as a compensatory mechanism. Second, the ADP400 might directly affect adipose tissue function, altering adiponectin secretion. This could occur through changes in the expression of genes involved in the synthesis or release of adiponectin.”
6. It has earlier been shown that adiponectin promotes the inhibitory phosphorylation of GSK3B at Ser 9 leading to suppression of Tau hyperproliferation. Given that Tau hyperphosphorylation is significantly high in ADP400, authors could have tested GSK3B ser 9 phosphorylation status instead of the total protein.
Response: We agreed with reviewer’s comments. Unfortunately, we did not measured p-GSK3β protein expression usimg Western blotting. We have examined the p-GSK3β/GSK3β protein expression in another study (manuscript in preparation), as expected, we did observe increased p-GSK activity in association with enhanced protein levels of p-Tau in the hippocampus.
7. Authors can cite PMID 38280832, which is a recent review summarizing the role of adiponectin in AD in both preclinical and clinical studies
Response: The paper is cited as suggested. In line 56.
References:
The role of adiponectin in Alzheimer's disease: A translational review Louise Sindzingre 1, Elodie Bouaziz-Amar 2, François Mouton-Liger 3, Emmanuel Cognat 4, Julien Dumurgier 4, Agathe Vrillon 4, Claire Paquet 4, Matthieu Lilamand 5
J Nutr Health Aging. 2024 Mar;28(3):100166.

Reviewer 2 Report
Comments and Suggestions for Authors
In the manuscript “Blocking Adiponectin Signaling Induces Alzheimer’s Disease-Like Neuropathology and Impairs Hippocampal function”, the authors proved that ADP400-treated mice exhibited impaired memory performance and increased anxiety-like behaviors. Adiponectin signaling may play roles in AD pathology by heightened hyperphosphorylation of tau, increased β-amyloid levels and decreased expression of PP2A in the hippocampus. ADP400 treatment significantly reduced hippocampal adult neurogenesis, as indicated by decreased BrdU, Ki67, and DCX staining. Meanwhile, inhibiting adiponectin signaling could lead to tau hyperphosphorylation and accumulated β-amyloid levels. This work provides correlation between adiponectin signaling and Alzheimer’s disease-like damages that may be contribute to build new mechanism and potential strategy.
1, Did the authors detect adiponectin signaling in the model of Alzheimer’s disease?
2, Whether or not did blocking adiponectin signaling result in metabolism-related effects? Is there any effect on other adiponectin-related signals?
3, Please explain how adiponectin signaling modulates Tau, amyloid, GSK and PP2A? Regarding these key proteins, did the authors perform immunostaining detection?
Author Response
Response to comment from Reviewer 2
1, Did the authors detect adiponectin signaling in the model of Alzheimer’s disease?
Response: We did not measure adiponectin signaling in AD model, however it is repeatedly found that deficiency of adiponectin increase risk of AD and AD-related neuropathology including increased p-tau and β-Amyloid levels. A review paper summarizing the important role of adiponectin in AD has been added in line 56.
2, Whether or not did blocking adiponectin signaling result in metabolism-related effects? Is there any effect on other adiponectin-related signals?
We have added the following sentences to the discussion: lines 329 - 338
“Blocking of adiponectin receptors with ADP400 could lead to significant metabolism-related effects, as adiponectin plays a crucial role in regulating metabolic processes [12]. Adiponectin enhances insulin sensitivity, so blocking its receptors (AdipoRs) could reduce insulin sensitivity, potentially contributing to insulin resistance and increasing the risk of type 2 diabetes. Disruption of adiponectin signaling might impair glucose uptake and utilization, leading to elevated blood glucose levels. Additionally, adiponectin influences lipid metabolism by promoting fatty acid oxidation and reducing triglyceride levels. Blocking AdipoRs could result in dyslipidemia, characterized by increased triglycerides and altered lipid profiles. Chronic disruption of adiponectin signaling through AdipoR blockade can have significant effects on metabolism and related pathways, potentially contributing to various metabolic and cardiovascular disorders. Further research is needed to fully understand the metabolic implications of ADP400 treatment.”
3, Please explain how adiponectin signaling modulates Tau, amyloid, GSK and PP2A? Regarding these key proteins, did the authors perform immunostaining detection?
Response: We have added in a new paragraph to describe the possible mechanisms of adiponectin on reducing p-tau with its interaction to GSK and PP2A. as below Line 301-315
“Although the underlying mechanisms of adiponectin’s action on modulating AD neuropathology remains largely un-clear, its signaling pathways can influence key proteins and enzymes involved in AD. Emerging findings suggest that adiponectin signaling might modulate p-tau accumulation through direct action on tau phosphorylation or indirect interactions with GSK3β and enzyme PP2A. PP2A dephosphorylated tau at several phosphorylation sites GSK3β is a crucial kinase that phosphorylates tau, contributing to accumulation of p-tau. Adiponectin signaling can inhibit GSK3β activity [4]. It attenuates tau hyperphosphorylation at multiple AD-related sites by activating Ser9-phosphorylated GSK3β with increased the Akt and PI3K activity [44]. Another study showed that adiponectin blocks the phosphorylation of the GSK-3 β / β-catenin pathway [45]. Additionally, adiponectin may enhance the activity of PP2A, a major phosphatase that dephosphorylates tau. Our unpublished data indicate that adiponectin knockout reduces PP2A activity and increases tau phosphorylation. Protein Phosphatase Methyl Esterase-1 (PME-1) is an enzyme that demethylates the catalytic subunit of PP2A, directly regulating its activity. This suggests that PME-1 may play a role in mediating the effects of adiponectin on the dephosphorylation of tau proteins [46,47]. Overall, adiponectin signaling may exert neuroprotective effects by modulating these pathways, potentially offering therapeutic benefits in AD. However, the exact mechanisms and the extent of these effects are still under investigation, and more research is needed to fully understand these interactions.”

Reviewer 3 Report
Comments and Suggestions for Authors
Dear Editor,
Thank you for the submitted manuscript, "Blocking Adiponectin Signaling Induces Alzheimer’s Disease-Like Neuropathology and Impairs Hippocampal Function" by Hui-hui Guo et al. The study discusses the effects of blocking adiponectin signaling in a mouse model and its implications for Alzheimer’s disease-like pathology. After carefully reviewing the manuscript, I have a few concerns and clarifications that I would appreciate the authors addressing:
- Several studies have previously demonstrated that blocking adiponectin results in Alzheimer’s disease-like pathologies, including hippocampus-related behavioral deficits and hyperphosphorylated tau accumulation. Some of these studies include:
- Ng RC et al., 2016 – Chronic adiponectin deficiency leads to Alzheimer’s disease-like cognitive impairments and pathologies through AMPK inactivation and cerebral insulin resistance in aged mice
- Min Woo Kim et al., 2017 – Suppression of adiponectin receptor 1 promotes memory dysfunction and Alzheimer’s disease-like pathologies
I would like to understand why these relevant citations were not included in the manuscript. - In the introduction (line 51), the citation style currently includes author names (e.g., “research by Ng RC et al.”). Kindly revise this to include only reference numbers in all instances.
- In the introduction (line 55), citation no. 17 refers to elevated adiponectin protein levels, but the manuscript mentions mRNA. Please verify and correct if necessary.
- In the introduction (line 80), the phrase "adiponectin signaling" seems to be an overstatement, as the study primarily presents phenotypic changes rather than directly analyzing signaling pathways. Kindly rephrase this for accuracy.
- In the methods section (line 97), could the authors clarify the rationale behind selecting the specific peptide treatment duration?
- Regarding Figure 1, I appreciate the inclusion of individual data points in the figures. However, based on the qPCR data for PP2A and GSK-3 beta, only 2 out of 4 cultures showed increased levels in the control group. Could the authors explain how data from only two cultures sufficiently supports the conclusion that ADP400 treatment decreases PP2A and GSK-3 beta?
- Regarding Figure 2:
a) Please separate B amyloid and beta-actin into distinct panels. These images contain 10 lanes, whereas the other markers have 8 lanes, making interpretation unclear. Additionally, kindly provide full blot images for all markers.
b) Many markers appear overexposed in the provided images. Could the authors supply lower-exposure data? For instance, tubulin appears highly saturated, and GSK-3 beta is cropped too much, leading to signal loss. - In Figure 2, is the observed increase in phospho-tau specific to the hippocampus? Did the authors examine phospho-tau levels in other brain regions?
- Regarding Figure 5, given the observed reduction in newborn neurons following ADP400 peptide treatment, did the authors assess any changes in the total neuronal population? Were additional neuronal markers evaluated? If not, I suggest including a comparison of neuronal marker expression between control and ADP400-treated groups.
- In the conclusion (line 324), the statement appears to be an overstatement given the data presented. Please consider rewording for accuracy.
- In the conclusion (line 358), there is no clear focus on adiponectin signaling. Did the study examine AMPK levels? If not, kindly revise the statement accordingly.
- Overall, the significance of the study is not entirely clear. Given that previous research has already established a connection between adiponectin signaling and Alzheimer’s disease pathology, could the authors better highlight the novel aspects of their findings?
Author Response
Response to comments from Reviewer 3
- Several studies have previously demonstrated that blocking adiponectin results in Alzheimer’s disease-like pathologies, including hippocampus-related behavioral deficits and hyperphosphorylated tau accumulation. Some of these studies include:
- Ng RC et al., 2016 – Chronic adiponectin deficiency leads to Alzheimer’s disease-like cognitive impairments and pathologies through AMPK inactivation and cerebral insulin resistance in aged mice
- Min Woo Kim et al., 2017 – Suppression of adiponectin receptor 1 promotes memory dysfunction and Alzheimer’s disease-like pathologies
I would like to understand why these relevant citations were not included in the manuscript. - In the introduction (line 51), the citation style currently includes author names (e.g., “research by Ng RC et al.”). Kindly revise this to include only reference numbers in all instances.
Response: Thanks for the contributive suggestions. We revised the citation style accordingly. We have included these two references in the introduction part in line 56-58
“Adiponectin deficiency in adiponectin knockout mice or suppression of adiponectin receptor 1 with shRNA approach leads to AD-like neuropathologies and memory dysfunction [19].”
3. In the introduction (line 55), citation no. 17 refers to elevated adiponectin protein levels, but the manuscript mentions mRNA. Please verify and correct if necessary.
Response: corrected with deleting extra wording “mRNA”
4. In the introduction (line 80), the phrase "adiponectin signaling" seems to be an overstatement, as the study primarily presents phenotypic changes rather than directly analyzing signaling pathways. Kindly rephrase this for accuracy.
Response: We have corrected the wording “adiponectin signaling” to adiponectin receptors.
5. In the methods section (line 97), could the authors clarify the rationale behind selecting the specific peptide treatment duration?
Response: This was a sub-chronic treatment with duration commonly seen for altering adult hippocampal neurogenesis. Since we examined changes of hippocampal structural and behavioral changes, we therefore adopted 3-week continuous treatment. We have mentioned in the materials and methods as below Line 107-108
“treatment, continuously for 21 days, a sub-chronic treatment duration for studying c a sub-chronic treatment duration for studying changes of hippocampal structural changes and behavioral changes”
6. Regarding Figure 1, I appreciate the inclusion of individual data points in the figures. However, based on the qPCR data for PP2A and GSK-3 β, only 2 out of 4 cultures showed increased levels in the control group. Could the authors explain how data from only two cultures sufficiently supports the conclusion that ADP400 treatment decreases PP2A and GSK-3 beta?
Response: The data with n=4 animal per group did not show significant differences between control and ADP400 treatment group. We have revised the description for this part accordingly as shown be below: line 299-300
“Our results demonstrated a decreasing trend in both PP2A and GSK3β mRNA expression in ADP400-treated mice. “
Regarding Figure 2:
a) Please separate B amyloid and beta-actin into distinct panels. These images contain 10 lanes, whereas the other markers have 8 lanes, making interpretation unclear. Additionally, kindly provide full blot images for all markers.
b) Many markers appear overexposed in the provided images. Could the authors supply lower-exposure data? For instance, tubulin appears highly saturated, and GSK-3 beta is cropped too much, leading to signal loss.
Response: (a) We have revised the figure to make it more clear, please see revised Figure 2.
(b) We agreed that the blotting images of the internal control tubulin for measuring p-tau levels were overexposed. There is clear significant difference between control and ADP400 treatment group. We apologies that we can’t provide images with less exposure time for the internal control alpha-tubulin. We have provided all raw Western blotting images as requested for the manuscript submission.
7. In Figure 2, is the observed increase in phospho-tau specific to the hippocampus? Did the authors examine phospho-tau levels in other brain regions?
Response: We did not measure different regions since the current study is focused on the hippocampus which is the key brain region being affected by AD and primary region for study memory dysfunction with alteration in adult neurogenesis. We will consider to extend our study in other regions such as prefrontal cortex in the future study.
8. Regarding Figure 5, given the observed reduction in newborn neurons following ADP400 peptide treatment, did the authors assess any changes in the total neuronal population? Were additional neuronal markers evaluated? If not, I suggest including a comparison of neuronal marker expression between control and ADP400-treated groups.
Response: We did not measure changes of neural population in the current study since we focus on the effects of ADP400 on changing adult-born neurons in the hippocampus, which is an important indicator for decreased hippocampal structural plasticity. Decrease neuronal population in the hippocampus. We will consider this in the future study with a research scope focusing on neuronal death and neuronal loss in chronic treatment with ADP400.
9. In the conclusion (line 324), the statement appears to be an overstatement given the data presented. Please consider rewording for accuracy.
Response: We have revised the conclusion statement as below:
“ Our study uncovered a significant effect of ADP400 peptide administration on increasing depression/anxiety-like behavior and impairing memory performance in concurrent with aberrant adult hippocampal neurogenesis in mice. The results suggest that pharmacological blocking adiponectin receptors could lead to AD-like neuropathology.”
10. In the conclusion (line 358), there is no clear focus on adiponectin signaling. Did the study examine AMPK levels? If not, kindly revise the statement accordingly.
Response: We have revised our wording to adiponectin receptor thorough the manuscript.
11. Overall, the significance of the study is not entirely clear. Given that previous research has already established a connection between adiponectin signaling and Alzheimer’s disease pathology, could the authors better highlight the novel aspects of their findings?
Response: We have added new content to highlight the novelty of our current study.
“Our results demonstrated that chronic intraperitoneal (i.p.) administration of ADP400 downregulated adiponectin receptor expression in the hippocampus. Adiponectin receptors are also present in other tissues, including the liver, muscles, heart, adipose tissue, pancreas, and kidneys. There is currently a lack of evidence demonstrating that ADP400 can cross the blood-brain barrier and enter the brain, we cannot rule out the possibility that the effects of i.p. administration of ADP400 are linked to changes in peripheral adiponectin secretion or peripheral inflammatory responses, which may lead to alterations in neuroinflammation. This suggests that the observed effects might not be limited to the downregulation of adiponectin receptor-mediated signaling in the brain. In conclusion, our study provides evidence for the detrimental effects of blocking adiponectin receptors by i.p. administration with adiponectin receptor antagonist, highlighting its negative effects in interrupting adiponectin receptors in the hippocampus, and its impact on tau hyperphosphorylation, anxiety-like behavior, cognitive function, and hippocampal neurogenesis.”

Round 2
Reviewer 3 Report
Comments and Suggestions for Authors
Dear Editor,
Thank you for providing the revised version of the manuscript. I appreciate the authors’ responses and the efforts made to address the comments. However, there are a couple of important points that still require clarification:
-
Unaddressed Comment from Previous Review
The authors did not address the first comment from my previous review. Several studies have demonstrated that blocking adiponectin leads to Alzheimer’s disease-like pathologies, including hippocampus-related behavioral deficits and hyperphosphorylated tau accumulation. Notably, the following studies provide relevant insights:- Ng RC et al., 2016 – Chronic adiponectin deficiency leads to Alzheimer’s disease-like cognitive impairments and pathologies through AMPK inactivation and cerebral insulin resistance in aged mice.
- Min Woo Kim et al., 2017 – Suppression of adiponectin receptor 1 promotes memory dysfunction and Alzheimer’s disease-like pathologies.
I would appreciate clarification on why these relevant citations were not included in the manuscript.
-
Clarifications Regarding Figure 5
Given the observed reduction in newborn neurons following ADP400 peptide treatment, did the authors assess any changes in the total neuronal population? Were additional neuronal markers evaluated? If not, I strongly recommend including a comparison of neuronal marker expressions between control and ADP400-treated groups. This is critical for the study’s conclusions.
If blocking adiponectin signaling results in reduced hippocampal function and cognitive impairments, the authors should provide data on the total neuronal population. The presence or impairment of adult neurogenesis does not necessarily indicate that the neurons are fully mature and functional. Therefore, I suggest including a figure with a mature neuronal marker to strengthen their findings.
Additionally, given that the quality of the western blotting data with tubulin is suboptimal, it is difficult to confirm whether there is a change in the neuronal population. Housekeeping proteins such as GAPDH and Actin are not neuron-specific, so their use as loading controls does not conclusively address this concern.
Thank you.
Author Response
Response to reviewer 3’s comments:
- Unaddressed Comment from Previous Review
The authors did not address the first comment from my previous review. Several studies have demonstrated that blocking adiponectin leads to Alzheimer’s disease-like pathologies, including hippocampus-related behavioral deficits and hyperphosphorylated tau accumulation. Notably, the following studies provide relevant insights: - Ng RC et al., 2016 – Chronic adiponectin deficiency leads to Alzheimer’s disease-like cognitive impairments and pathologies through AMPK inactivation and cerebral insulin resistance in aged mice.
- Min Woo Kim et al., 2017 – Suppression of adiponectin receptor 1 promotes memory dysfunction and Alzheimer’s disease-like pathologies.
I would appreciate clarification on why these relevant citations were not included in the manuscript.
Response:
We have cited these two references in the manuscript in the previous revised version, please see line 56-58 in Page 2.
“Adiponectin deficiency in adiponectin knockout mice or suppression of adiponectin receptor 1 with shRNA approach leads to AD-like neuropathologies and memory dysfunction [19,20].”
- Clarifications Regarding Figure 5
Given the observed reduction in newborn neurons following ADP400 peptide treatment, did the authors assess any changes in the total neuronal population? Were additional neuronal markers evaluated? If not, I strongly recommend including a comparison of neuronal marker expressions between control and ADP400-treated groups. This is critical for the study’s conclusions.
If blocking adiponectin signaling results in reduced hippocampal function and cognitive impairments, the authors should provide data on the total neuronal population. The presence or impairment of adult neurogenesis does not necessarily indicate that the neurons are fully mature and functional. Therefore, I suggest including a figure with a mature neuronal marker to strengthen their findings.
Response:
The current study mainly focused on changes of hippocampal plasticity, adult neurogenesis is well known be one of the structural plasticity changes critically involved in regulation of memory performance. We examined three different markers that indicate neuronal survival (BrdU+ve cells), cell proliferation (Ki67+ve cells) and immature neurons (doublecortin+ve cells), we have studied three different important stages of adult neurogenesis process as indicators for changes in adult neurogenesis. We highly appreciate reviewer’s comment on quantifying total neuronal population. However, quantifying total neuronal population will not be able to reflect changes in neuronal maturation and function. Mature neuronal marker like NeuN stains all population of neurons including exiting neurons born in developmental stage and adult born neurons, therefore quantification of total neuronal population using neuronal marker will be studying neuronal loss rather than change in neuronal maturation of newborn neurons. We have addressed reviewer’s suggestion as a future study in discussion part: see lines 357-360
“ Our results have demonstrated significant reduction in survival rate of newborn neurons, number of proliferating cells and immature neurons by ADP400, future study addressing changes in total neuronal populations in the hippocampus warrant further investigation to examine if ADP400 induces neuronal loss or impairs neuronal functions of hippocampal adult-born neurons.”
Additionally, given that the quality of the western blotting data with tubulin is suboptimal, it is difficult to confirm whether there is a change in the neuronal population. Housekeeping proteins such as GAPDH and Actin are not neuron-specific, so their use as loading controls does not conclusively address this concern.
Response:
Our western blot experiment aimed to examine changes of p-Tau, β-amyloid protein expression levels in the hippocampal tissues, but not to examine changes of neuronal population.
Housekeeping proteins are often used as internal references or loading controls to normalize the expression of the protein of interest. They are also used as normalization references in molecular biology research. Some common housekeeping proteins used for Western blots include: β-actin, β-tubulin, GAPDH. Based on our experimental aim, we followed the basic principles for selection of housekeeping protein in Western blot analysis include (1) The housekeeping protein should not be neuronal specific and should not be affected by treatment condition, (2) The protein level of the loading control shouldn't change based on experimental conditions. (3) The loading control and the protein of interest should be easy to tell apart on the blot.
Round 3
Reviewer 3 Report
Comments and Suggestions for Authors
Thank you for providing the revised version.
-
For the authors’ reference, the two citations (references 19 and 20) have been included only in the current revision. The claim that these references were present in the previous version is incorrect. We have thoroughly reviewed both the manuscript and the response. If this was an oversight, we kindly request the authors to acknowledge it rather than redirecting the issue to the reviewers.
-
Regarding the total neuronal population, if the authors strongly assert that altered neurogenesis is a key finding in this manuscript, we request a high-quality image of DCX staining in the ADP 400-treated group. The current image appears faint and seems to reflect expression levels rather than the actual neuronal count.
-
We acknowledge the fundamental principles of housekeeping genes and their use in experiments. However, the housekeeping signals in the provided Western blot data appear oversaturated, and low-exposure data have not been included. Additionally, the loading volumes seem inconsistent across multiple wells. Given these concerns, it remains unclear how the data can be accurately analyzed with saturated blot signals.
Author Response
Thank you for your valuable comments. Please see the attachment.
